

# Neighbourhood and stand structure affect stemflow generation in a heterogeneous deciduous temperate forest

Johanna C. Metzger[1,2], Jens Schumacher[3], Markus Lange[2], Anke Hildebrandt[1,2,4,5]

[1]Institute of Geosciences, Friedrich Schiller University, Burgweg 11, 07749 Jena, Germany
[2]Max-Planck-Institute for Biogeochemistry, Hans-Knöll-Straße 10, D-07745 Jena, Germany
[3]Institute of Mathematics, Friedrich Schiller University, Ernst-Abbe-Platz 2, 07743 Jena, Germany
[4]Helmholtz Centre for Environmental Research – UFZ, Permoserstraße 15, 04318 Leipzig, Germany
[5]German Centre for Integrative Biodiversity Research (iDiv), Deutscher Platz 5e, 04103 Leipzig, Germany

*Correspondence to*: Johanna C. Metzger (johanna.clara.metzger@uni-jena.de)

**Abstract.** Although stemflow oftentimes represents only a small portion of net precipitation in forests, it creates hot spots of water input that can affect subsurface stormflow dynamics. The distribution of stemflow over different trees is assumed to be temporally stable, yet often unknown. Therefore, it is essential to know the systematic factors driving stemflow patterns. Several drivers have been identified in the past, mainly related to tree traits. Less attention has yet been paid to tree neighbourhood interactions impacting stemflow generation and creating stand patches with enhanced or reduced stemflow.

We recorded stemflow in 26 precipitation events on 65 trees, growing in 11 subplots (100 m² each), in a temperate mixed beech forest in the Hainich National Park, Germany. We used linear mixed effects models to investigate how traits of individual trees (tree size, tree species, number of neighbouring trees, their basal area, and their relative height) affect stemflow and how stemflow is affected by stand properties (stand, biomass and diversity metrics).

As expected, stemflow increased with event and tree size. Stemflow was highly variable at both tree and subplot scale.

Especially in large rainfall events (> 10 mm), tree/subplot ranking was almost identical between events, probably due to fully developed flow paths bringing out the full stemflow potential for each tree. Neighbourhood and stand structure were increasingly important with event size (15 % of fixed effects on the tree scale, ca. 65 % on the subplot scale for large events). Subplot scale stemflow was especially enhanced by a higher proportion of woody surface, expressed by a high number of trees, low leaf area and a large maximum tree size. Simpson's diversity index contributed positively to stemflow yield in large

events, probably by allowing more efficient space occupation. Also, our models suggest that neighbourhood impacts individual tree morphology, which may additionally increase stemflow in dense, species diverse neighbourhoods. Unexpectedly, rain shading within the canopy had little impact on stemflow spatial variation.

Overall, we find a strong cross-scale temporal stability. Tree size and tree density were the main drivers, independently increasing stemflow, creating forest patches with strongly enhanced or reduced stemflow. Our results show that, besides tree

metrics, also forest structure and potentially diversity affect stemflow patterns and associated potentially biogeochemical hotspots.



## 1 Introduction

In forests, precipitation is intercepted by the canopy and reaches the soil partitioned into throughfall and stemflow. The different pathways of precipitation through the forest canopy create a strongly heterogeneous pattern of water input to the soil, with consequences for soil hydrobiochemistry (Levia & Frost, 2003; Zimmermann et al., 2007). They compartmentalize the forest floor into cold and hot spots of infiltration, with a strong subsequent impact on subsurface flow and biogeochemical processes (Liang et al., 2007; Guswa & Spence, 2012; Coenders-Gerrits et al., 2013). Thus, understanding of forest canopy precipitation partitioning processes is highly important for our conceptual understanding of forest ecohydrology systems.

Although stemflow constitutes a minor fraction of net precipitation, research shows that stemflow is important for a site's hydrological functioning (Pressland, 1976; Durocher, 1990; Levia & Frost, 2003; Hildebrandt et al., 2007; Staelens et al., 2008; Levia & Germer, 2015). Stemflow introduces a strong additional heterogeneity to subcanopy precipitation. Stems potentially act as funnels and can make trees prominent hot spots of canopy drainage. Concentrated water inputs to the soil can trigger macropore flow (Flühler, 2004), bypassing the soil and thresholding subsurface stormflow processes that contribute to deep percolation (Taniguchi et al., 1996; Liang et al., 2007). This effect has been called double-funneling of trees (Martinez-Meza & Whitford, 1996; Johnson & Lehmann, 2006; Li et al., 2009; Liang et al., 2011; Schwärzel et al., 2012) and renders trees important players in the hydrological functioning of forests, on top of their role for soil water depletion.

Correctly depicting variance of stemflow and understanding its mechanisms can be of utter importance, because according to the hot spots and hot moments hypotheses (McClain et al., 2003), maximum fluxes have the strongest impact on the system (rather than site averages). However, only few studies provide measures of stemflow spatial (i.e., tree-to-tree) variation (Hanchi & Rapp, 1997; Zimmermann et al., 2015). Most stemflow studies focus on few trees to exemplary encompass a site's stemflow processes. This is probably because elaborate sampling is required to capture stemflow variance: A random and representative sample is needed, encompassing a high coverage and extent within the study stand. The limited data that are available show that that stemflow variation is substantial, and higher for stemflow compared to throughfall (Metzger et al., 2017, van Stan et al., sub). Thus, stemflow contributes importantly and even primarily to net precipitation heterogeneity and flux hot and cold spots and moments. At the same time, some research suggests that tree-to-tree stemflow variation is stable in time (Metzger et al., 2017), meaning that at different precipitation events, the same trees produce relatively high or relatively low stemflow. Although few studies have explicitly investigated this temporal stability in stemflow, a great deal of research has been conducted to link tree traits to stemflow yield to understand spatial (i.e., tree-to-tree) variability, and thus inherently implied temporally stabile drivers. Most prominently, tree diameter (or circumference, basal area) has been identified as factor shaping tree-specific stemflow within an event (Reynolds & Henderson, 1967; André et al., 2008; Krämer & Hölscher, 2009; Takahashi et al., 2011). Yet, stemflow yield still shows a great deal of between-tree variation after accounting for tree size (Reynolds & Henderson, 1967; Takahashi et al., 2011, McKee & Carlyle-Moses, 2017), as trees traits related to morphology and crown architecture additionally affect tree-individual stemflow. For example, many and steeply inclined branches (Herwitz, 1987; Návar et al., 1993; Iida et al., 2005; Levia et al., 2015, Martinez-Meza & Whitford, 1996) smoother bark (Aboal et al., 1999;



Iida et al., 2005; van Stan & Levia, 2010; van Stan et al., 2016), leaf hydrophobicity (Iida et al., 2005), low LAI / few leaves (Takahashi et al. , 2011; Molina & del Campo, 2012; Levia et al., 2015) and more woody surface (Levia & Germer, 2015; Levia et al., 2015) been found to enhance stemflow production.

So far, most of the investigations have targeted species specific variables. However, some canopy traits are also affected by
stand structure: Trees have been shown to strongly adapt their growth to space occupation of neighboring trees (Schröter et al., 2012; Juchheim et al., 2017). Different competition strategies and typical phenotypes of different species complement each other in mixed forest, allowing a more efficient niche and space occupation (Frech et al., 2003; Juchheim et al., 2017). Thus, also neighborhood characteristics such as species composition, diversity or size heterogeneity could impact tree traits related to stemflow. Stand and neighborhood properties might directly and indirectly influence stemflow formation of the individual
tree. This pattern could also persist on a larger scale, forming forest patches of structure-induced enhanced and reduced stemflow.

Yet, neighborhood effects have hardly been considered for stemflow analyses. Some studies have included canopy position, (Reynolds & Henderson, 1967; Aboal et al., 1999; Terra et al., 2018) or neighboring tree proximity (McKee & Carlyle-Moses, 2017) in tree stemflow models, while Krämer & Hölscher (2009) tested species composition effects on area average stemflow.
Other studies have discussed a shading effect in the lower canopy (André et al., 2008; Takahashi et al. 2011) as possible explanation for their stemflow results. However, a systematic study explicitly focusing on neighborhood effects on stemflow in a quantitative approach is yet missing.

In this contribution we tackle spatiotemporal patterns of stemflow in conjunction with spatially distributed tree and neighborhood variables using a spatially stratified design. Additionally, by assessing stemflow area-based on 11 small (100
m²) subplots, we obtain a first assessment of impacting effects on areal integrated stemflow patterns at the subplot scale.

Based on the above design, we assess the temporal stability of spatial stemflow patterns, test the impacts of stand structure and neighborhood parameters (additional to tree size) on individual stemflow yield and whether these factors even out for stemflow variation at a larger scale.

## 2 Methods

### 2.1 Site description and sampling design

The measurement site is situated in a gently sloping forested area in the Hainich low mountain range in central Germany. It is a site of the Hainich Critical Zone Exploratory within the Collaborative Research Center AquaDiva (Küsel et al., 2016). Annual rainfall ranges between 600 mm and 900 mm. The mixed beech forest is part of a National Park and is unmanaged, with a high age and species heterogeneity. Within a 1 ha sampling plot, 11 subplots of 10 m × 10 m each were chosen in a regular pattern
and stemflow measured on all trees within the subplots (Fig. 1). 65 trees were such selected, 80 % of which are beech trees (Fagus sylvatica), 12 % sycamore maple (*Acer pseudoplatanus*). *Acer platanoides*, *Fraxinus excelsior*, *Carpinus betulus* and *Ulmus glabra* also occurred. Diameter at breast height (DBH) shows a negative exponential distribution, with 54 % of the trees



having a DBH < 0.1 m and a maximum DBH of 0.81 m. Tree metrics within the subplots are representative for the larger scale stand (see Appendix).

## 2.2 Field sampling

### 2.2.1 Measurement of stand properties

Trees within the plot and a 10 m buffer zone around it were surveyed and given and identification number (ID). The position of each tree was determined using a total station (Topcon, Tokyo, Japan) combined with a differential GPS (Topcon, Tokyo, Japan). Tree height was measured using an ultrasonic sensor (Haglöf Vertex, Haglöf, Järfälla, Sweden) and tree circumference at breast height was measured with a measuring tape in 2014. The trees' DBH and basal area were calculated from their circumference at breast height, assuming a circular shaped tree trunk. Leaf area index was measured in summer 2015 by a
LAI-2000 (LI-COR, Lincoln, Nebraska USA).

### 2.2.2 Neighborhood of individual trees

From the stand properties we derived metrics describing the neighborhood of the 65 individual trees. As neighborhood, we defined an area around the tree with a radius of the mean tree distance on the plot:

$$r = 2\sqrt{\frac{\frac{A_{\text{plot}}}{n_{\text{trees,plot}}}}{\pi}},\tag{1}$$

where $r$ is the mean tree distance, which equals 4.7 m on our plot, $A_{\text{plot}}$ is the plot area and $n_{\text{trees,plot}}$ is the total number of trees on the plot. Within this radius, we counted the number of trees in the neighbourhood, calculated their cumulative basal area, and the neighbourhood's relative height ($h_{\text{n,rel}}$), as follows:

$$h_{\text{n,rel}} = \frac{h_{\text{tree,max}}}{h_{\text{tree},j}},\tag{2}$$

where $h_{\text{tree,max}}$ is the height of the highest tree in the neighbourhood and $h_{tree,j}$ is the height of reference tree $j$. Note that relative
height, as a neighbourhood property, increases for taller neighbourhoods.

### 2.2.3 Subplot characteristics

We calculated heterogeneity measures for each stemflow subplot. We used Simpson's index of biodiversity $D$ (Simpson, 1949), as it is suitable for small sample sizes, $-\log_e D$ transformed, as recommended by Buckland (2005):

$$D = \sum_i \left(\frac{n_i}{\sum_i n_i}\right)^2,\tag{3}$$

where $n_i$ is the number of individuals of species $i$ per unit area.

Additionally, we derived a size heterogeneity index $H$, which was calculated according to Krämer & Hölscher (2009) as:



$$H = \frac{\frac{q_{0.75,s} - q_{0.25,s}}{q_{0.5,s}}}{\tilde{s}},  \qquad (4)$$

with

$$s = h_{\text{tree}} d_{\text{tree}},  \qquad (5)$$

Where $q_{x,s}$ is the x[th] quantile of $s$ and $h_{tree}$ and $d_{\text{tree}}$ are the height and DBH of a tree.

**2.2.4 Gross precipitation and Stemflow measurement**

Gross precipitation and stemflow were measured as described in Metzger et al. (2017). For gross precipitation, five funnel-type collectors were used, which were placed ca. 250 m from the forest plot on an adjacent grassland, ca. 50 m away from the forest edge. Precipitation in mm was derived referring the precipitation volume in the collectors to the area covered by the funnel and taking the median of the five parallel measurements.

Stemflow was collected on all trees within the eleven designated subplots (see above) into containers by way of collars made from lay-flat hose wrapped around the trees and sealed with silicone. Precipitation was sampled at an event basis from May to August in the years 2014, 2015 and 2016, recording all occurring events. Sampling started ca. 2 h after the event ended. Measurements lasted several hours. If measurements were interrupted by new rainfall, events were treated cumulatively. Over the entire period a total 39 events were recorded. Events, where overflow of containers could have occurred for at least one stemflow measurement, were excluded from the data analysis. For the statistical model analysis (see below), we also excluded very small events (< 0.5 L median stemflow per tree), leaving 26 of the 39 sampled precipitation events. Subplot stemflow was calculated as the sum of stemflow collected from all trees on that subplot.

The stemflow funnelling ratios were calculated from the individual stemflow volumes as follows (Herwitz, 1987):

$$R_{\text{F}} = \left( \frac{V_{\text{SF}}}{P_g \cdot A_{\text{tree}}} \right),  \qquad (6)$$

where $R_{\text{F}}$ is the funnelling ratio, $V_{\text{SF}}$ the stemflow volume, $P_g$ the gross precipitation and $A_{\text{tree}}$ a tree's basal area. It shows, to what degree a tree concentrates the rainfall to a point water input to the soil.

Normalized tree/subplot stemflow $V_{SFn}$ was calculated from the tree individual stemflow volume ($V_{\text{SF},j,e}$) for event $e$, and the event's median stemflow volume ($\widetilde{V_{SF,e}}$), according to Vachaud et al. (1985):

$$V_{SFn} = \left( \frac{V_{\text{SF},j,e} - \widetilde{V_{SF,e}}}{\widetilde{V_{SF,e}}} \right),  \qquad (7)$$



### 2.3 Statistical analysis

#### 2.3.1 Descriptive statistics of stemflow patterns

To examine temporal stability of stemflow patterns, we correlated individual/subplot stemflow yields pairwise for all events falling into an event size class, thus obtaining a set of correlation coefficients for each event size class. In order to account for

non-normal distribution of stemflow, we used Spearman's rank correlations. Sets of high (or low) correlation coefficients thus signify that the same (or not) trees/subplots produce above and below average stemflow yields in each event, demonstrating high (or low) temporal stability.

#### 2.3.2 Linear mixed effects models

In order to determine the effect of potential driving factors for stemflow yield, linear mixed effects models (LMM) were used.

LMM are multivariate linear regression models which allow to control for repeated sampling. Quantified factors, the impact of which is to be tested in the model, are called fixed effects. Qualitative information of repeated sampling, referring to individuals, time points or treatments, are called random effects. Random effects can explain parts of the residual of the fixed effects model by calculating different intercepts for different category levels. In a random slope model, random effect category levels can also change the slopes of the linear regression of certain fixed effects (so-called interactions). In this way, repeated

sampling cannot bias the fixed effects models. R (R core team, 2016) was used for all data processing and analysis. Linear mixed effects models were developed using the packages lme4 (Bates et al., 2015) and lmerTest (Kuznetsova et al., 2016), pseudo R² were calculated using the package MuMIn (Barton et al., 2018).

We developed models at two spatial scales: (1) tree-individual scale and (2) aggregated subplot scale, in both cases assessing how precipitation, tree size and neighbourhood affect stemflow. For (1) we fitted $P_g$, tree DBH, tree height, neighbourhood

number of trees, neighbourhood basal area and neighbourhood relative height as fixed effects and precipitation event ID, event year, tree ID, tree species and subplot ID as random effects. For (2) we fitted $P_g$, as well as the number of trees, number of species, Simpson's diversity index, stand basal area, maximum DBH, size heterogeneity and LAI on the subplot as fixed effects and precipitation event ID, event year and subplot ID as random effects. Table 1 and Table 2 summarize of the fixed and random effects of both models. We grouped measured precipitation events into size classes (small: < 3 mm, medium: 3−10

mm, large: > 10 mm) similarly as in Metzger et al. (2017). Because of the exclusion of events < than 0.5 L median stemflow, less events representative of the small and medium size class were left in our data set, and we expanded the range for the small events class to 5 mm, yielding 5 small, 7 medium and 16 large events. Thus, at each scale (tree individual and subplot), four linear effects models were developed, three for the individual event size classes and one including all events.

#### 2.3.3 Data selection and transformation for linear mixed effects models

All data was checked for normal distribution and was log-transformed if necessary (stemflow volumes and tree DBH). To be able to account for values zero of stemflow, 1 was added to the stemflow data before transformation. All data was standardized





automatically using the "scale" function in R. This normalization allows assessing the single effects' impacts by comparing the slopes (fixed effects) and intercepts (random effects) fitted for each factor. All tested metrics are listed in Table 1 (fixed effects) and Table 2 (random effects).

### 2.3.4 Model development

The model development involved the improvement of the mixed effects model by optimizing or excluding effects until only significant effects remain and the model has a low error. This was done successively by repeated comparison of two models which differ in one aspect only and choosing the significantly better one in terms of the AIC (Akaike information criterion). The model development was here conducted in two main steps: (1) Development of the random effects model: Starting with a complete model including all possible fixed and random effects, the significance of random effects was tested separately for

each effect. Hereby, selection started with the effect with the highest standard deviation, testing all possible interactions, the simple effect (no interaction) and exclusion of the effect. Only significant random effects were retained. (2) Development of the fixed effects model from the established random effects model. Hereby, selection started with the effect with the lowest slope estimate, testing whether the model improved significantly by inclusion of the effect. Only significant fixed effects were retained.

## 3 Results

### 3.1 Event and stemflow characteristics

We recorded 38 precipitation events with total $P_g$ (gross precipitation) of 626 mm (Table 3). Roughly half of the events fell into the class "large" ($P_g$ > 10 mm). Overall, only a small fraction of rainfall (1.8 %) was converted to stemflow, but the contribution changed with event size (Fig. 2). Small and medium events (50 % of the events) only contribute 4 % of total

stemflow in our study area. Most of the stemflow (96 %) was derived from events classified as "large". Moreover, 80% of the stemflow are generated in the 30 % largest events. Also, 30 % of the total measured stemflow was generated in one single large precipitation event of 65 mm (30 May 2014).

Event funnelling ratios increased with event size (Fig. 3, left) from a median of 1 for small events to a median of 7 for medium events and to a median of 14 for large events. Maximum values range from 60 for events with rainfall < 30 mm to over 200

for the largest recorded event with 65 mm. As funnelling ratios increase with event rainfall, local input near stems increases relative to gross precipitation with event size. Thus, large events not only contribute most to total stemflow, but additionally enhance the funnelling effect. Non-beech trees on our plot are on average as productive as the beech trees (Fig. 3, right).

The coefficient of quartile variation (CQV) for all events averaged out at 0.65, for large events it increased to 0.7. Between subplots, variation for all events as well as for large events amounted to 0.55 (Fig. 4, left).

Spatial patterns of stemflow were temporally stable (Fig. 4, right, Fig. 5, Fig. 6). This is especially true for large rainfall events (Fig. 4, right). The median correlation coefficient between stemflow in events of the large event class is 0.9 and is significantly





($p \ll 0.001$) higher than in small or medium events both on the tree and the subplot scale. This indicates that on both scales systematic drivers of stemflow are active for large events. Additionally, higher ranks in stemflow did not always correspond to those in DBH (Fig. 5 & Fig. 6).

**3.2 Site, vegetation and neighbourhood factors affecting stemflow**

**3.2.1 Tree individual models**

All linear mixed effects models for tree individual stemflow cover much of the variation in observed stemflow yields ($R^2 = 0.77$–0.91, Table 4). Yet, for medium events, most of the variance is explained by the random effects, which implies that the non-measured individual and site properties had an overall big effect on stemflow, while included factors were not as important.

Considering modelled fixed effects, as expected, event rainfall ($P_g$) is the most important and significant effect in all event size classes (Table 4). For small and medium events, $P_g$ explains most (99 % and 83 %) of stemflow in the fixed effects. However, for large events, $P_g$ is less important while tree size (e.i. DBH) becomes more important: 48 % of stemflow is explained by $P_g$ and 37 % is explained by DBH in the fixed effects in large events.

Neighbourhood properties (number of trees, basal area or relative height) have a significant impact on stemflow for the small

events, and they are a trend ($p = 0.077$ and $0.055$) in medium and large events. Which neighbourhood parameters are important varies with event class, while the direction of the effects (i.e. increasing or decreasing stemflow) is consistent in all event classes. Neighbourhood effects increase with event size from small to large events, while at the same time, gross precipitation decreases from small to large events. Neighbourhood properties thus affect stemflow comparatively stronger in large events. In large events, the number of trees in the neighbourhood increases stemflow, while stemflow is decreased by a larger basal

area and taller trees in the neighbourhood. Overall, neighbourhood "crowding" (i.e. parameters indicating high biomass, like neighbourhood basal area) tends to decrease stemflow production per tree with one notable exception: Neighbourhood's number of trees increases stemflow yield.

Additional neighbourhood effects may be hidden in the random effects which encompass unquantified but systematic effects of repeated measurements within a group or individual. Of those, subplot ID is almost never significant (Table 4). Instead,

event ID is the strongest random effect for all models, accounting for rain event characteristics not captured by total event rainfall. The interaction with tree diameter shows that the prominent relationship between tree diameter and stemflow changes with the individual event properties. The second strongest effect is tree ID, acting as proxy for tree parameters other than those quantified in the fixed effects, e.g. tree morphological features. Interaction of $P_g$ with tree ID indicates that individual trees may yield more or less stemflow, depending on the event precipitation. Further, tree species is a significant random effect only

for large events, interacting with DBH, showing that the relation between DBH and stemflow is species-specific. Event year only appears in the model for medium sized events with a very small contribution. Overall, the random effects reflect the





substantial importance of tree properties other than DBH for generating stemflow, specifically tree individual morphology and position (tree ID) and tree species.

### 3.2.2 Subplot scale models

All mixed effects models for subplot stemflow explain a large proportion of variance, higher than for the tree individual models above ($R^2$ = 0.85–0.95, Table 5). Similar to the tree individual models, in medium events, the random effects have a higher share in the explanation of variance than the fixed effects.

$P_g$ and the number of trees on the subplot are the most important fixed effects (Table 5). Their relative contribution shifts from small and medium events to large events, with $P_g$ loosing and number of trees gaining importance (ca. 95 % and 5 %, 75 % and 15 %, 15 % and 20 %). For all event sizes, $P_g$, number of trees and maximum DBH increase stemflow, while subplot basal area, LAI and most of the diversity measures (both number of species and size heterogeneity index) decrease it. Exception is the Simpson's species diversity index, which also increases subplot stemflow.

Only one random effect, event ID, is significant for all subplot models (Table 5). Neither event year nor subplot ID played a role for any of the models, indicating that plot properties were sufficiently captured by the fixed effects. This is further supported by the high proportion of fixed effects contributing to the explained variance, specifically in large events ($R^2$ = 0.93, thereof 0.74 for the fixed- and 0.19 for the random effects model, Table 5).

### 3.2.3 Comparison of tree and subplot scale models

In both the individual and subplot scale, the model encompassing all events was dominated by the random effects, although both in small and large events most of the variance was explained by (different) fixed effects. This shows that driving factors differ between event size classes, and we will therefore focus mainly on event class models.

Generally, R² values are higher for the subplot than for the tree individual model. The subplot-scale model thus was better able to explain the data variation. Moreover, R² of the fixed effects are higher on the subplot scale and R² of the random effects (as well as the model residual within the random effects) was higher in the tree individual model.

The regression slopes between predicted and observed data are slightly smaller than 1 at both scales, indicating a bias towards underestimation (see example for large events in Fig. 7). The model bias of the subplot model (slope of 0.92) is lower than that of the tree individual model (0.87). Consequently, when calculating subplot stemflow from tree individual model predictions, the prediction bias is slightly worse (slope of 0.9) than that of the subplot level model itself (Fig. 7). The same procedure allows evaluating the role of tree ID at the subplot scale. Remember that tree ID in the tree individual models could potentially include neighbourhood effects, specifically morphology (enhancing individual stemflow without effect on neighbor) or shading (enhancing individual stemflow at the expense of the neighbour). For this, we calculated subplot sums of stemflow predicted by the tree individual model with and without including the tree ID random effect in the model. The regression slope for the prediction without tree ID was only 0.86 (vs. 0.9 with tree ID included, see Fig. 7). The difference is




not significant but a trend, showing that tree ID contributes to increasing stemflow in one (or several) individuals on the subplot without decreasing it in others.

In general, similar patterns emerge for different event size classes on the tree scale and on the subplot scale: $P_g$ is a strong driver for stemflow at both scales and loses influence with increasing event size, yet more so at the plot scale. Instead, tree or stand characteristics affect stemflow, especially in large events. On both the tree individual and subplot scale, absolute tree size and the number of trees most strongly increase stemflow, while neighbourhood/subplot basal area slightly decreases stemflow. Species become relevant on both scales especially for large events. Event ID is the strongest random effect on both scales, while subplot ID was not significant as a random effect on either scale.

Yet, we also observe small differences between the tree individual and subplot scale model patterns: For tree individual models, apart from $P_g$, individual tree size is most important in large events and neighbourhood effects play a minor role. In contrast, for the subplot model, several stand structural parameters affect stemflow. Especially, the number of species and the number of trees are more important than $P_g$ and tree size. Notably, while the size heterogeneity index on the subplot scale significantly decreases stemflow in large events, we found no effect of the equivalent measure (relative height) on the individual tree scale.

## 4 Discussion

Stemflow varied substantially in space both on the tree individual as well as on the subplot scale. At the same time the greatest share of stemflow volume is created during large events, when spatial patterns of stemflow are particularly temporally stable, both on the tree individual as well as on the plot scale. This shows that besides throughfall, the temporal stability of which has been repeatedly reported, also stemflow patterns are equally or even more stable in time (Metzger et al., 2017). Also, funnelling ratios increased with increasing event size. Our findings confirm that (i) spatial patterns in stemflow are systematic and therefore can be explained by tree or stand properties, which we try to identify in this study, and (ii) large events generate the majority of total stemflow, have the highest funnelling ratios, and spatial patterns are the most pronounced and stable.

### 4.1 Tree size affects stemflow only in large events with fully developed flow paths

Tree metrics are the most important fixed effects in large events (but is less important in small events), which is likely related to the establishment of fully connected stemflow paths. Fully connected flow paths lead to the built-up of stable, systematic patterns of stemflow and increased funnelling ratios, relating strongly to tree properties. This agrees with previous research on stemflow generation processes: Although some studies conceptualized stemflow invoking a bucket concept, where tree (André et al., 2008) or bark (Aboal et al., 1999) storage need to saturate before stemflow is initiated, a more dynamic picture is given by Herwitz (1987), Crockford et al. (1996) and Levia & Frost (2003), which fits well with our observation. They say that "stemflow generation can begin before the woody frame is completely wetted" (Levia & Frost, 2003) due to preferential flow lines resulting from tree morphology or angled rain. One step further go Levia et al. (2011) and Van Stan et al. (2016), describing the development of new flow paths with progressing rainfall time, as additional tree surfaces get wetted. In either





of these cases, stemflow generation depends on critical event size thresholds. This view is supported by our findings: At small events, factors shaping spatial stemflow patterns are mostly random and of low temporal stability, indicating that flow paths are not yet well established. Medium events are characterized by increased temporal stability of spatial ranks, however low explained variance in the fixed effects, indicating that flow paths are only partly developed. For large events, tree traits related to water collection or channelling capability are the most important factors explaining individual tree stemflow, which indicates that flow paths are fully established. Together, these results suggest that increasingly established flow paths with increasing event size invoke spatially stable patterns of stemflow that are more related to tree attributes and less to event properties.

### 4.2 Neighbourhood and stand properties affect stemflow

### 4.2.1 Stand structure effects largely explain subplot stemflow

For large events on the subplot scale, all proposed stand structural parameters are significant. Subplot ID has no random effect, thus, selected stand characteristics in the fixed effects capture the stemflow generation processes on the subplot scale well, including also those unexplained morphological factors which are hidden in the tree ID on the individual tree scale. Also, the subplot scale model explains more variance compared to the tree individual model.

For large events on the tree individual scale, in neighbourhood effects only appeared only as trends, which may have been related to different neighbourhood variables, such as number of trees vs. basal area, working in different directions. However, the subplot models reveal that those neighbourhood effects identified at the individual tree level act in the same way at the subplot level: The number of trees still increases the stemflow on the subplot level, while basal area reduces it. This shows that a tree's neighbours systematically affect its stemflow and that those patterns are not cancelling each other out when considering community stemflow at the subplot scale. Moreover, this suggests that the tree morphologic properties hidden in tree ID on the tree individual scale are actually shaped by stand and neighbourhood dynamics. In conclusion, neighbourhood effects were better covered by subplot properties than by the metrics of the individual neighbourhood. Accordingly, knowledge of stand structure proves to be advantageous for stemflow assessment.

### 4.2.2 Tree density positively affects stemflow, shading plays a subordinate role

Number of trees is the most prominent positive contributor to stemflow on the subplot level, confirming the intuitive rule that more trees produce more stemflow. Similarly, Reynolds & Henderson (1967) found higher interception in denser stands, which potentially turns into stemflow after a rainfall threshold. Accordingly, Molina & Del Campo (2012) report increased stemflow for higher stand densities. Levia & Frost (2003), Levia & Germer (2015) and Levia et al. (2015) repeatedly argue that more woody surface area (hit by raindrops and providing stemflow pathways) is a main prerequisite for enhanced stemflow. This implies that - next to bigger trees or trees with more branches - also a higher number of trees potentially increase stemflow. Interestingly, the number of trees in the neighbourhood also increases individual tree stemflow, which is far less intuitive than the equivalent at the subplot scale. The number of neighbours could also enhance a tree's stemflow by promoting steeper





branching angles in dense stands (Schröter et al., 2012; Juchheim et al.,2017), which are known to yield more stemflow (Návar et al., 1993; Levia et al., 2015, see below). Molina & del Campo (2012) similarly observed increased stemflow production in denser stands also at the individual tree scale in a Mediterranean climate but attributed the effect to evaporation protection under dense canopies, as they varied density in their study by thinning and could so exclude canopy morphology as reason.

Alternatively, dripping on smaller trees may contribute to stemflow generation (see below).

Additional to higher tree density, also reduced leaf area increased stemflow, potentially by increasing the exposed woody surface. This agrees with former studies on the effect of tree properties on stemflow generation (Van Stan & Levia, 2010; Takahasi et al., 2011; Molina & del Campo, 2012; Levia et al., 2015). Rain intercepted by leaves is rather redirected away from the stem and becomes throughfall, as leaves are not steeply inclined toward the branch, especially when they are wet.

The most frequently proposed direct neighborhood impact in literature is a rain shading effect, where exposed canopies collect more precipitation than less exposed ones (Takahashi et al., 2011; Santos Terra et al, 2018). André et al. (2008) discussed that small trees overtopped by larger neighbors might be deprived of a great part of rainfall. Similarly, amongst others (Crockford & Richardson, 1990; Návar et al., 1993; Aboal et al., 1999), Levia & Frost (2003) found higher stemflow production in the upper canopy. Yet, in Reynolds & Henderson (1967), medium height, co-dominant and subdominant trees were most efficient

in stemflow production. Pointing in the same direction, smaller trees are often reported to have higher stemflow funnelling ratios (Murakami, 2009; Van Stan & Levia, 2010), and our data support this.

Relative height as a fixed effect was never significant in our models. In contrast, the combination of number and size (basal area) of neighbouring trees impact a single tree's stemflow. Our data also suggest, that the highest or largest tree does not coercively yield the most stemflow (tree height was not retained in the tree scale model and ranks of DBH and stemflow yield are not the same). Highest trees are best competitors for light, which implies tree traits which are not beneficial for stemflow

production: Small crowns, few branches and a low DBH per height ratio (Juchheim et al., 2017). Also, thick leaf layers in the light canopy could divert rainfall from the tree, as a high LAI reduces stemflow production (see above).

In conclusion, stemflow is enhanced by tree density, only limited by trade-offs between trees when basal area increases. We thus find the positive impact of tree density much stronger compared to the shading effect between trees, which, in contrast, is

much weaker than expected.

### 4.2.3 Neighbourhood influences stemflow indirectly by shaping tree morphology

Apart from the neighbourhood effects revealed by those factors characterizing the neighbourhood, as discussed above, there is a "dark figure" of potential neighbourhood interactions hidden in the random effects on the tree scale, specifically event year, tree ID, and subplot ID. The year of the measurement covers canopy dynamics as growth and canopy gaps due to windfall

and broken branches, changing both the tree and its neighbourhood. Subplot ID stands for the properties of the small tree community the respective tree is situated in and which are not covered by the fixed effects describing the neighbourhood. Tree ID comprises all kinds of tree traits (canopy architecture) and canopy position effects (shading or exposure) which are not covered by the fixed effects.



Of those random effects, tree ID is the most prominent significant random effect in all event classes. Interestingly, when predicting subplot scale stemflow using the individual scale model for large events, the subplot stemflow is underestimated, and more so than by predicting the subplot stemflow using the subplot scale model. Therefore, the tree ID-induced variance on the tree scale does not cancel out on the subplot scale. This further supports the conclusion that interactions are not shading,

but more likely stemflow-enhancing tree morphology effects.

Neighbourhood impacts stemflow indirectly, as it shapes the growth of a tree's canopy (Schröter et al., 2012; Juchheim eta l., 2017) and stands representative for small tree communities, as species and ages do not mix randomly, but appear in clusters. At the same time, the morphology of a tree substantially affects stemflow: Aboal et al. (1999) found, that bigger crown projection area, the position in the canopy and smoother bark yielded higher stemflow volumes. Návar et al. (1993) reported

higher stemflow yields for trees with many, steeply inclined branches from the top part of the crown. Iida et al. (2005) attributed branching angles to changes in precipitation partitioning and more branches and thus higher crown length to higher stemflow. In a study on beech saplings, Levia et al. (2015) identified, from a set of properties, besides woody surface, more and steeper branches and fewer leaves as significantly promoted stemflow.

Since every tree is a dynamic imprint of its direct environment, neighbourhood and its temporal development drive a tree's

traits. Our results suggest that this reflects on stemflow yield. Additional measurements of canopy architecture would be required to confirm potential effects of stand density on tree morphology in our plot.

### 4.3 Tree diversity increases stemflow, possibly due to effective canopy space occupation

Most of the parameters capturing diversity and heterogeneity of the stand decrease stemflow, with the notable exception of the Simpson's index. This may be related to the fact that our forest plot is beech dominated, where at the same time, fully grown

beech trees produce a great deal of stemflow (André et al., 2008; Krämer & Hölscher, 2009; Van Stan & Levia, 2010).

Our results are in line with observations by Krämer & Hölscher (2009), who found a decrease in stemflow with species diversity (Shannon index) in a nearby forest and attributed this result to the high beech proportion on their site as a strong driver for stemflow. Schroth et al. (1999) also observed reduced stemflow in mixed stands, yet they argued that this finding would strongly depend on the involved species and their traits

However, in forest stands dominated by stemflow prolific tree species, increasing stand heterogeneity implies both a decrease in tree size and introduces less stemflow producing species. Thus, heterogeneity measures need to be interpreted with caution, especially when measurements on representative trees are used.

The parameter "number of species" reflects rather a reciprocal of the number of large beech trees on the subplot than a measure of species richness. This is because most trees (80 %) are beech and the number of species is strongly related to the number of

small trees (DBH ≤ 0.11 m, $R^2 = 0.88$) on the subplots. Also, size heterogeneity reduces stemflow generation in medium and large events. Stronger size heterogeneity implies the coexistence of both very large and very small individuals, where, in terms of stemflow, the smaller individuals potentially add little to the effect of the prolific large tree(s).



Furthermore, Juchheim et al. (2017) showed a significant change in beech morphology when mixed with other species, of a kind potentially enhancing stemflow. Therefore, intermixture of other tree species in beech dominated forests may have a positive impact on stemflow production, specifically for the beech trees, but not necessarily for the intermixed non-beech trees. Notably, the Simpson's index at the subplot scale is positively related to stemflow. Simpson's index is a relative measure of species diversity that corrects for the number of individuals considered (Buckland, 2005). The Simpson's index illustrates not the mere number of species, but balanced species abundance, and is therefore sensitive for the strong beech dominance we find on most subplots (Magurran, 2004). The Simpson's index significantly increases stemflow only for large events, where flow paths are established, and individual tree trait effects on stemflow develop their full potential. Frech et al. (2003) have shown that more diverse tree communities are very efficient in using the canopy space. As different species follow different strategies to compete for resources, they form variable canopy shapes which makes it easier for trees of different species to move closer together. A more efficient occupancy of canopy space increases woody surface area, the existence and exposition of which has been shown to be the core of stemflow promotion (see above, Levia & Frost, 2003; Levia & Germer, 2015; Levia et al., 2015). Additionally, beech trees in mixture with other species are more likely to develop crown morphologies with a higher number of branches (Juchheim et al., 2017), thus, further promoting stemflow (Levia et al., 2015).

## 5 Conclusion

In this study, we investigated possible neighbourhood effects on stemflow yield on the individual tree and subplot (patch) scale. Our unmanaged and mixed species forest produced a high spatial variance in tree individual stemflow. Spatial patterns of stemflow were temporally stable, especially for large events. The spatial variance persisted with the same order of magnitude on small forest patches of $10 \times 10$ m.

Tree size was not the only relevant trait for stemflow generation. Neighbourhood and stand properties contributed importantly to stemflow distribution. On both investigated scales, stemflow increased with the number of trees in the neighbourhood. Tree density especially increases woody surface area – a key to stemflow promotion, providing rain receiving area and flow paths. Because neighbourhood effects did not cancel out on the subplot scale, tree morphology (crown architecture) must have enhanced subplot stemflow. As canopies react plastically towards their surroundings, neighbourhood impacts tree morphological features, including those affecting stemflow. In contrast, shading within the canopy was much less important: Relative height did not affect stemflow, only neighbourhood and stand basal area, representing larger trees, slightly reduced stemflow which suggests a weak shading effect. Also, barely decreased stemflow variance at the subplot scale indicates that shading effects are probably minor.

All impacts are most obvious for large precipitation events. Tree, stand and neighbourhood effects are more important as event size increases. We conclude that the full development and connection of drainage flow paths through the canopy taps the full potential of systematic factors in forest structure impacting stemflow yield. Because of positive effects on forest density, unmanaged and mixed species forest could be more stemflow-productive than managed ones. This is supported by the positive



effect of Simpson's diversity index on small stand stemflow. More research required to understand systematic effects of forest management on stemflow.

**Data availability**

Data will be provided via a data repository.

**Author contribution**

JM and AH designed the experiments. JM carried them out with the assistance of AH and students named in the Acknowledgements. JS mentored and counselled the linear mixed effects modelling and its interpretation. ML mentored and counselled the linear mixed effects model interpretation and the use of pseudo R². JM developed the model code. JM prepared the manuscript under guidance by AH and with contributions from all co-authors.

**Competing interests**

AH is part of the editorial board of the journal. Otherwise, the authors have no competing interests.

**Acknowledgements**

This study is part of the Collaborative Research Centre AquaDiva (CRC 1076 AquaDiva) of the Friedrich Schiller University Jena, funded by the Deutsche Forschungsgemeinschaft. We thank the Hainich CZE site manager Robert Lehmann and the

Hainich National Park. Field work permits were issued by the responsible state environmental offices of Thüringen.
We thank our technician Janett Filipzik and the Bachelor, Master and Ph.D. students Lisa-Marja Ahrens, Marcel Bechmann, Fabio Brill, Nicolas Dalla Valle, Malte Kleemann, David Kunze, Ricardo Ontiveros-Enriquez, Martin Roggenbuck, Danny Schelhorn, Josef Weckmüller and Sven Westermann for their contributions to the data collection.

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



**Tables**

**Table 1: Distributed parameters of tree, neighborhood and subplot properties used as fixed effects in the linear mixed effects models of the named scale.**

| Fixed effect parameter | | Used on scale | Median | IQR | Maximum | Minimum |
|---|---|---|---|---|---|---|
| **Tree (n = 65)** | **DBH [m]** | tree | 0.11 | 0.22 | 0.81 | 0.05 |
| | **height [m]** | tree | 16.0 | 13.4 | 36.2 | 4.5 |
| **Neighborhood (n = 65, 70 m²)** | **# trees** | tree | 4 | 6 | 16 | 0 |
| | **basal area [m²]** | tree | 0.17 | 0.37 | 0.64 | 0.00 |
| | **relative height** | tree | 1.55 | 1.64 | 6.84 | 0.00 |
| **Subplot (n = 11, 100 m²)** | **# trees** | subplot | 5 | 2 | 21 | 2 |
| | **# species** | subplot | 1 | 1 | 5 | 1 |
| | **Simpson's index** | subplot | 0.00 | 0.47 | 0.82 | 0.00 |
| | **basal area [m²]** | subplot | 0.28 | 0.41 | 0.83 | 0.02 |
| | **maximum DBH [m]** | subplot | 0.55 | 0.26 | 0.81 | 0.10 |
| | **size heterogeneity index** | subplot | 1.37 | 0.79 | 16.84 | 0.56 |
| | **LAI** | subplot | 3.93 | 0.66 | 4.95 | 3.40 |

Abbreviations: DBH: Diameter at breast height, #: Number of, LAI: Leaf area index



**Table 2: Type, number and range of values of categorical variables used as random effects in the linear mixed effects models on the named scale.**

| Random effect parameter | | Used on scale | Number of levels | Levels |
|---|---|---|---|---|
| **Event** | **ID** | tree & subplot | 26 | Event identification number (1 − 26) |
| | **Year** | tree & subplot | 3 | 2014, 2015, 2016 |
| **Tree** | **ID** | tree | 65 | Tree identification number (1 − 65) |
| | **Species** | tree | 5 | *Acer platanoides,* *Acer pseudoplatanus,* *Carpinus betulus,* *Fagus sylvatica,* *Fraxinus excelsior,* *Ulmus glabra* |
| **Subplot** | **ID** | subplot | 11 | Subplot identification number (1 − 11) |

Abbreviations: ID: Identification number



**Table 3: Overview of collected stemflow precipitation events. Measured stemflow depth refers to cumulative stemflow of one event of all trees that could be evaluated. Events that were excluded from the linear mixed effects modeling are labeled and the reason for the exclusion given (see Methods section for more detail). Gap filled stemflow is only available for events included in the modeling analysis. The overall gap rate was 6.2 %, missing a mean of 5.2 % of the calculated total stemflow.**

| | Event properties | | | | Stemflow depth | | | |
| | | | | | Measured | | Gap filled | |
| ID | Date | $P_g$ [mm] | Size class | Excluded for | $P_{SF}$ [mm] | $P_{SF}/P_g$ [%] | $P_{SF}$ [mm] | $P_{SF}/P_g$ [%] |
|---|---|---|---|---|---|---|---|---|
| 21 | 6/14/2015 | 1.1 | small | median too low | > 0.01 | 0.01 | - | - |
| 17 | 5/10/2015 | 1.15 | small | - | > 0.01 | 0.03 | > 0.01 | 0.03 |
| 31 | 7/21/2015 | 1.57 | small | - | > 0.01 | 0.14 | > 0.01 | 0.14 |
| 25 | 6/28/2015 | 1.79 | small | median too low | > 0.01 | 0.08 | - | - |
| 23 | 6/20/2015 | 2.05 | small | median too low | > 0.01 | 0.08 | - | - |
| 9 | 6/5/2014 | 2.35 | small | median too low | > 0.01 | 0.20 | - | - |
| 19 | 5/30/2015 | 2.76 | small | median too low | 0.01 | 0.40 | - | - |
| 22 | 6/18/2015 | 3.31 | medium | median too low | 0.01 | 0.44 | - | - |
| 5 | 5/19/2014 | 3.66 | medium | median too low | 0.05 | 1.24 | - | - |
| 20 | 6/2/2015 | 3.71 | medium | - | 0.01 | 0.19 | 0.01 | 0.19 |
| 18 | 5/13/2015 | 4.09 | medium | - | 0.04 | 0.89 | 0.04 | 0.94 |
| 27 | 7/11/2015 | 4.58 | medium | median too low | 0.04 | 0.77 | - | - |
| 16 | 7/26/2014 | 4.69 | medium | - | 0.04 | 0.84 | 0.04 | 0.86 |
| 39 | 6/28/2016 | 5.27 | medium | - | 0.01 | 0.10 | 0.01 | 0.11 |
| 32 | 7/25/2015 | 5.66 | medium | - | 0.09 | 1.57 | 0.09 | 1.67 |
| 13 | 7/11/2014 | 6.31 | medium | - | 0.17 | 2.74 | 0.18 | 2.92 |
| 1 | 5/4/2014 | 8.24 | medium | - | 0.06 | 0.79 | 0.11 | 1.29 |
| 11 | 7/2/2014 | 10.3 | large | - | 0.04 | 0.42 | 0.05 | 0.46 |
| 10 | 6/11/2014 | 10.5 | large | - | 0.27 | 2.56 | 0.29 | 2.72 |
| 6_7 | 5/26/2014 | 11 | large | - | 0.23 | 2.09 | 0.23 | 2.13 |
| 26 | 7/8/2015 | 13.32 | large | - | 0.37 | 2.75 | 0.39 | 2.93 |
| 38 | 6/21/2016 | 13.68 | large | - | 0.13 | 0.94 | 0.13 | 0.94 |
| 28_29 | 7/15/2015 | 13.87 | large | - | 0.36 | 2.60 | 0.36 | 2.62 |
| 36 | 6/16/2016 | 16.92 | large | - | 0.17 | 1.01 | 0.19 | 1.10 |
| 43 | 8/2/2016 | 19.63 | large | - | 0.24 | 1.24 | 0.25 | 1.26 |
| 40 | 7/4/2016 | 19.79 | large | - | 0.17 | 0.88 | 0.17 | 0.88 |
| 33 | 7/28/2015 | 20.12 | large | - | 0.84 | 4.17 | 0.90 | 4.48 |
| 34 | 5/25/2016 | 20.8 | large | median too low | 0.49 | 2.36 | - | - |
| 24 | 6/24/2015 | 23.01 | large | median too low | 0.66 | 2.86 | - | - |
| 37 | 6/16/2016 | 23.15 | large | - | 0.31 | 1.33 | 0.31 | 1.33 |
| 41 | 7/14/2016 | 24.12 | large | - | 0.67 | 2.77 | 0.67 | 2.77 |





| 35 | 5/31/2016 | 25.02 | large | - | 0.66 | 2.64 | 0.70 | 2.79 |
| 42 | 7/25/2016 | 33.51 | large | median too low | 0.98 | 2.94 | - | - |
| 30 | 7/20/2015 | 35.19 | large | overflow | 1.79 | 5.07 | - | - |
| 15 | 7/23/2014 | 35.81 | large | - | 1.15 | 3.20 | 1.29 | 3.60 |
| 14 | 7/14/2014 | 42.24 | large | overflow | 0.91 | 2.15 | - | - |
| 8 | 5/30/2014 | 64.99 | large | - | 3.53 | 5.43 | 3.58 | 5.51 |
| 12 | 7/10/2014 | 86.8 | large | overflow | 3.69 | 4.25 | - | - |

Abbreviations: $P_g$: Gross precipitation, $P_{SF}$: Stemflow net precipitation



**Table 4: Results of the linear mixed effects models for individual tree stemflow yield: Slope estimates and significance levels of significant fixed effects, standard deviations of random effects and their interacting fixed effects (random slopes). The four models include (i) all precipitation events, (ii) small precipitations events with rainfall < 5 mm, (iii) medium precipitation events with rainfall 3 – 10 mm, (iv) large precipitation events with rainfall > 10 mm. Pseudo-R² are given for each full model (fixed and random effects),** 5 **for the fixed effects model separately and for the random effects model separately. Note that data was scaled before model development.**

| | | All events | Small events | Medium events | Large events |
|---|---|---|---|---|---|
| **R²** | | | | | |
| **Full model** | | 0.91 | 0.86 | 0.77 | 0.84 |
| **Fixed effects** | | 0.19 | 0.73 | 0.11 | 0.51 |
| **Random effects** | | 0.72 | 0.12 | 0.66 | 0.33 |
| **Relative effect size** | | | | | |
| **Fixed effects** | **Gross precipitation** | ↑ 0.28 *** | ↑ 7.72 *** | ↑ 1.04 *. | ↑ 0.28 *** |
| | **Tree DBH** (log.) | ↑ 0.25 *** | - | ↑ 0.17[1] | ↑ 0.22 * |
| | **Tree height** | - | - | - | - |
| | **Neighborhood # trees** | - | ↑ 0.1 ** | - | ↑ 0.05 . |
| | **Neighborhood basal area** | ↓ 0.05 ** | - | - | ↓ 0.04 . |
| | **Neighborhood relative height** | - | - | ↓ 0.05 . | - |
| **Random effects (interaction)** | **Event ID** | 0.68 (Tree DBH) | 0.24 (Tree DBH) | 0.40 (Tree DBH) | 0.17 (Tree DBH) |
| | **Event year** | 0.23 (N. # trees) | - | 0.04 (Tree height) | - |
| | **Tree ID** | 0.15 (Tree height) | 0.17 (-) | 0.41 (Gross precip.) | 0.16 (Gross precip.) |
| | **Tree species** | - | - | - | 0.06 (Tree DBH) |
| | **Subplot ID** | 0.09 (Gross precip.) | - | - | - |
| | **Residual** | 19.0 | 0.37 | 0.30 | 0.19 |

[1] Effect was not significant, but necessary for the model's convergence

Abbreviations: DBH: Diameter at breast height, log.: Log-transformed, #: Number of, n.: Neighborhood, precip.: Precipitation, rel.: Relative

Levels of significance: *** : p < 0.001, ** : p < 0.01, * : p < 0.05, . : p < 0.1



**Table 5: Results of the linear mixed effects models for subplot stemflow: Slope estimates and significance levels of significant fixed effects, standard deviations of random effects and their interacting fixed effects (random slopes). The four models include (i) all precipitation events, (ii) small precipitations events with rainfall < 5 mm, (iii) medium precipitation events with rainfall 3 – 10 mm, (iv) large precipitation events with rainfall > 10 mm. Pseudo-R² are given for each full model (fixed and random effects), for the fixed effects model separately and for the random effects model separately. Note that data was scaled before model development.**

| | | All events | Small events | Medium events | Large events |
|---|---|---|---|---|---|
| **R²** | | | | | |
| **Full model** | | 0.95 | 0.89 | 0.85 | 0.93 |
| **Fixed effects** | | 0.21 | 0.76 | 0.40 | 0.74 |
| **Random effects** | | 0.74 | 0.13 | 0.45 | 0.19 |
| **Relative effect size** | | | | | |
| **Fixed effects** | **Gross precipitation** | ↑ 0.33 *** | ↑ 7.44 *** | ↑ 2.03 * | ↑ 0.32 *** |
| | **# trees** (log.) | ↑ 0.30 *** | ↑ 0.42 *** | ↑ 0.43 *** | ↑ 0.42 *** |
| | **# species** | ↓ 0.13 ** | - | ↓ 0.23 *** | ↓ 0.50 *** |
| | **Simpson's index** | - | - | - | ↑ 0.23 ** |
| | **Basal area** | - | - | - | ↓ 0.13 ** |
| | **Maximum DBH** | ↑ 0.12 *** | - | - | ↑ 0.30 *** |
| | **Size heterogeneity index** (log.) | ↓ 0.08 *** | - | ↓ 0.06 * | ↓ 0.12 *** |
| | **LAI** | - | - | - | ↓ 0.07 *** |
| **Random effects (interaction)** | **Event ID** | 0.76 (# species) | 0.31 (# species ) | 0.36 (Simpson's in.) | 0.20 (Maximum DBH) |
| | **Event year** | - | - | - | - |
| | **Subplot ID** | - | - | - | - |
| | **Residual** | 0.21 | 0.34 | 0.21 | 0.12 |

Abbreviations: #: number of, log.: log-transformed, DBH: diameter at breast height, LAI: leaf area index, ID: identification number, in.: index

Levels of significance: *** : $p < 0.001$, ** : $p < 0.01$, * : $p < 0.05$, . : $p < 0.1$





**Figures**

**Figure 1: Position of the eleven subplots (grey shaded areas, 10 x 10 m) in which stemflow was sampled within the forest plot.**







Figure 2: Ranked cumulated subplot stemflow (bars) per event for each event size class (top: small, < 5 mm, middle: medium, 3–10 mm, bottom: large, > 10 mm) and the contributions of individual trees (alternating light and dark blue sections of each bar).





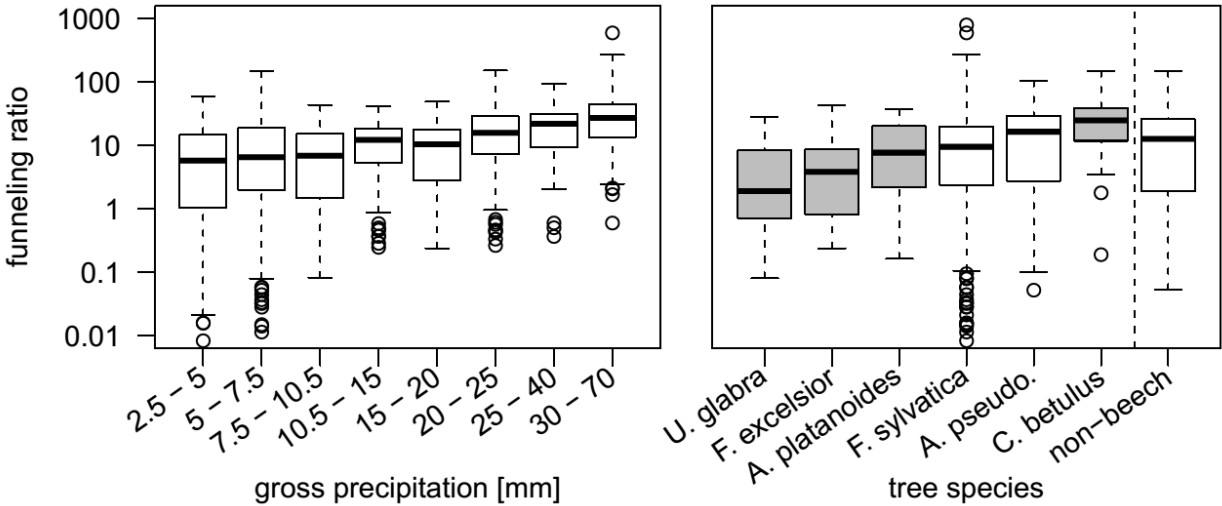

**Figure 3: Event funneling ratios of individual trees (n=65), (left) in relation to event gross precipitation, (right) in relation to tree species. Grey shaded boxplots contain the data of less than three tree individuals.**





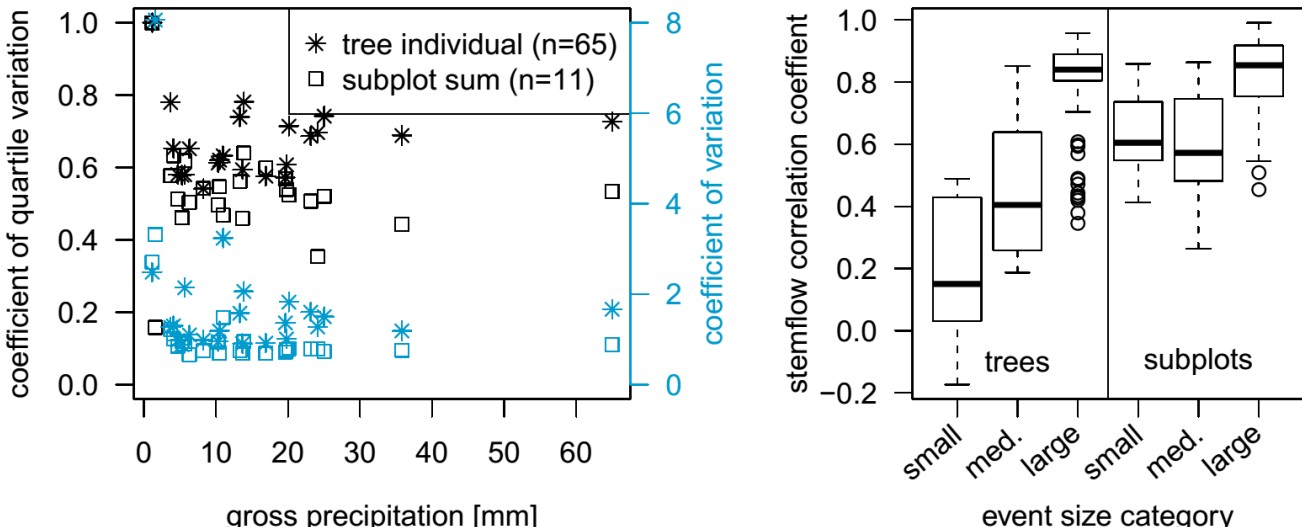

**Figure 4: (Left)** Coefficients of quartile variation and coefficients of variation for stemflow of individual trees and subplots for all recorded precipitation events in relation to gross precipitation. **(Right)** Temporal stability of stemflow on the tree and the subplot scale, calculated as pairwise correlation coefficients (Spearman) of tree/subplot stemflow between all different precipitation events of one event size class.





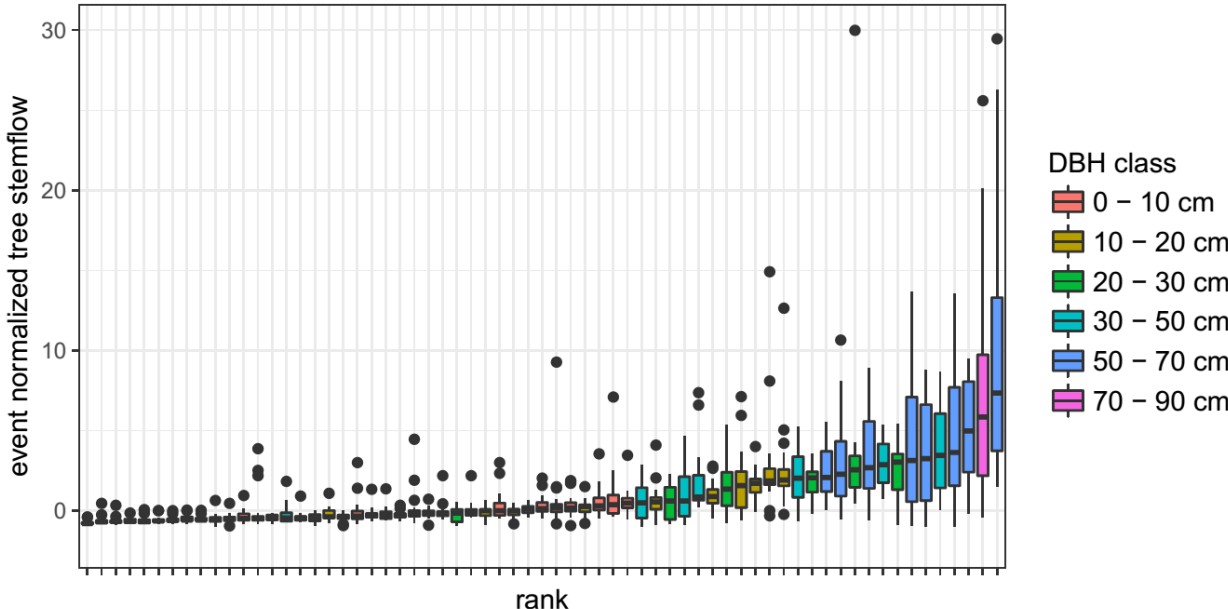

**Figure 5: Temporal stability of tree individual stemflow over all sampled events. Trees are ranked according to their median event normalized stemflow and colored according to DBH (diameter breast height).**





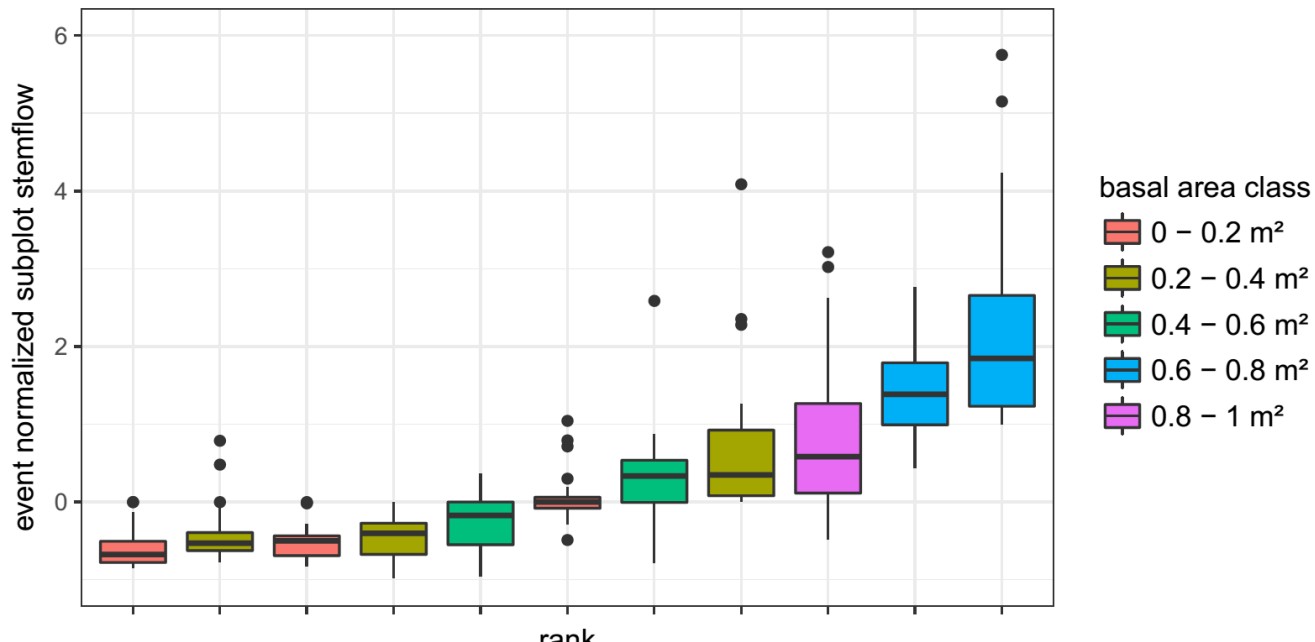

**Figure 6: Temporal stability of 100 m² subplot stemflow over all sampled events. Subplots are ranked according to their median event normalized stemflow and colored according to basal area.**





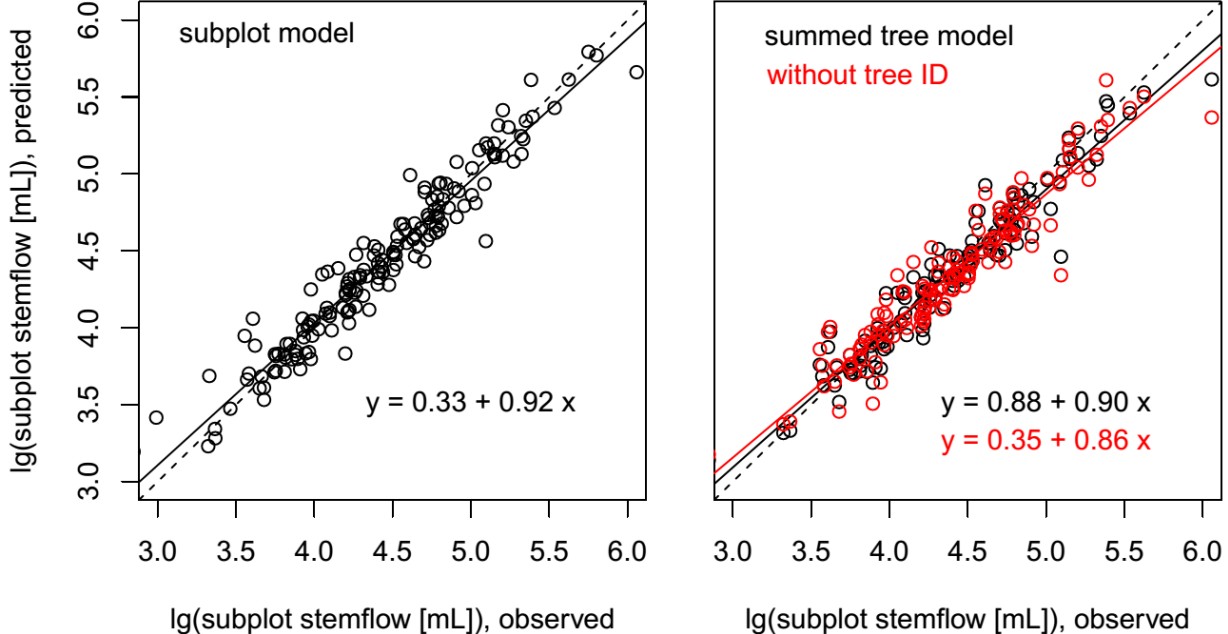

**Figure 7: (Left) Predicted stemflow per subplot using the subplot linear mixed effects model in relation to observed values for the large event class, (right) stemflow sums per subplot predicted by the individual tree linear mixed effects model in relation to observed values. The right panel shows additionally the predicted values when excluding the tree ID random effect from the individual tree model. Dashed lines give the 1-to-1-line, continuous lines show the linear regressions, equations are given in the graph.**





**Appendix**



**Figure A1: Histograms of stand properties on the whole 1-ha-plot (left, n = 581) and the eleven 100-m²-subplots on which stemflow was measured (right, n = 65).**



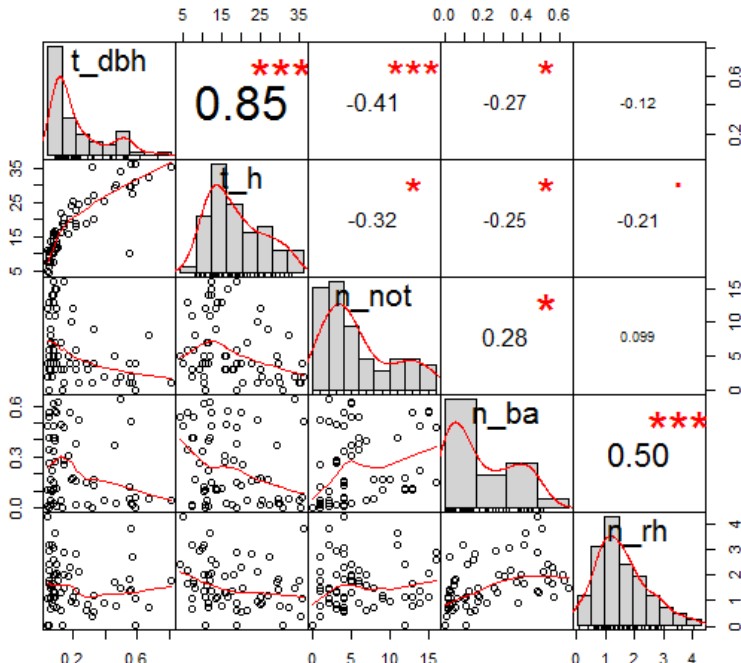

**Figure A2: Distributions and correlations of variables included as fixed effects in the linear mixed effect models of tree individual stemflow. Abbreviations: t_dbh: Tree DBH, t_h: Tree height, n_not: Number of trees in the neighborhood, n_ba: Neighborhood basal area, n_rh: Neighborhood relative height. DBH: Diameter at breast height.**

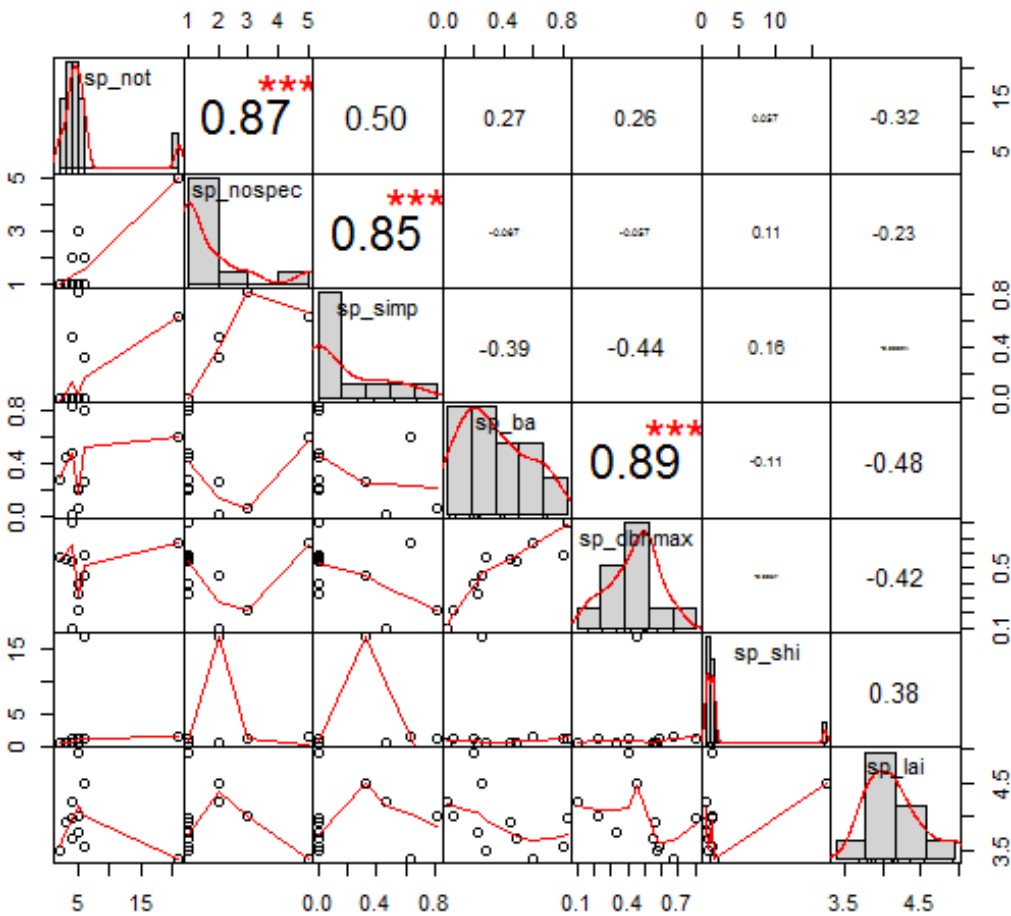

**Figure A3: Distributions and correlations of variables included as fixed effects in the linear mixed effect models of subplot stemflow. Abbreviations: sp_not: Number of trees in the subplot, sp_nospec: Number of species in the subplot, sp_simp: Simpon's diversity index of the subplot, sp_ba: Basal area of the subplot, sp_dbhmax: DBH of the biggest tree on the subplot, sp_shi: Size heterogeneity index of the subplot, sp_lai: Subplot LAI. DBH: Diameter at breast height, LAI: leaf area index.**