# Peer review of "Neighbourhood and stand structure affect stemflow generation in a heterogeneous deciduous temperate forest"

_Hydrology and Earth System Sciences, 2019_

## Referee Comment (RC1) · Anonymous Referee #1 · 28 Jul 2019

General Comments

The article by Metzger et al. was a pleasure to read. I found the paper to be novel, well written, well referenced, methodologically (field and statistical analyses) sound, and of wide appeal to a global audience. The tables and figures were also clear and readily interpretable. Therefore, I recommend publication following minor revision. My comments mostly regard readability and some minor technical comments that the authors should address.

Specific Comments

1) The statistical linear mixed effects models was nicely done. However, to make the

analysis readily accessible to the reader, I recommend a flow chart walking one through the analysis. Such a figure in the Methods section would really clarify the sequence of the analysis and serve the reader.

2) The last sentence of the abstract is a bit awkward to read and should be rephrased.

3) Section 4.1 might benefit from the incorporation of Levia et al 2010 (https://doi.org/10.1016/j.jhydrol.2009.10.028) as this paper specifically discusses tree size in relation to the development of stemflow at the intra-storm scale, even showing the delay in stemflow production by larger trees of the same species due to differences in bark water storage capacity.

4) Page 10, line 30: I believe that the authors are referring to the following reference: https://doi.org/10.1016/j.atmosenv.2011.03.022 and not the Levia et al (2011) reference currently cited. Please check.

5) Page 11, line 27: please delete "repeatedly"

6) Section 4.2.2: I really found this section interesting and I liked how some of the forest ecology aspects were brought in. In this light, I think that the authors would find the 1971 classic book "The Adaptive Geometry of Trees" by Henry Horn (Princeton University Press) of great interest. It is their choice as to whether they wish to add this text to their paper but I believe that they would find it both informative and useful.

---

## Referee Comment (RC2) · Anonymous Referee #2 · 6 Aug 2019

My comments and observations are inline with the comments posted by reviewer #1. The article by Metzger et al. was very well done. The methodology was sound and the paper sheds new and interesting light on factors influencing stemflow production that will appeal to a global audience. I also recommend publication following minor revision. My specific comments are as follows: Page 2, Line 4: Perhaps of starting the sentence with "They" the authors could say "These pathways..." Page 2, Lines 9 – 11. Perhaps the reference of Carlyle-Moses et al. 2018 (Advances in Water Resources) could be added here. Page 2 and 3, Lines 32 – 3: Crown projection area has also been found to be very, very important and probably should be included here. Page 7, line 21: Perhaps state: "...the largest 30 % of events" Page 14, Line 17: Perhaps "individual

tree stemflow" instead of "tree individual stemflow" Table 3: should the sign ">" be "<"
0.01 in the Psf column. I believe the authors mean less than and not more than here.

---

## Author Comment (AC1) · 7 Aug 2019

Many thanks to Referee #2 for the comments on our manuscript. We are very thankful for the general approval of the article as well as for the observant and helpful input for its improvement, including the mistakes found in Table 3. We agree with Referee #2 in all points and will implement the suggestions and corrections of the Specific Comments in our manuscript.

---

## Referee Comment (RC3) · Anonymous Referee #1 · 8 Aug 2019

Yes, the work flow diagram that you have developed is exactly along the lines that I was thinking. The clearer, the better. This will help readers follow the flow of the analysis.

---

## Author Comment (AC2) · 8 Aug 2019

We thank Referee #1 very much for the supportive and encouraging comment on our manuscript. We thank also for the attentive review and the suggestions in the Specific Comments, giving us the opportunity to further improve the article. We are currently working on their incorporation:

We made a flow chart of the model selection process, yet, it looked very complicated, which is why we switched to a workflow presentation instead (figure given below). We hope that Referee #1 finds the figure suitable for making the process easily accessible to the reader. If Referee #1 wants to give an opinion and/or improvement ideas on the

workflow, we would be glad to learn them.

We are also reviewing the literature Referee #1 suggested, including the book "The adaptive geometry of trees" by Henry Horn, which is ordered for interlibrary loan but has yet to be delivered.

[Figure]

[Figure]

**Fig. 1.**

---

## Author Comment (AC3) · 16 Aug 2019

We are happy about Referee #1 approving of the workflow we presented in reply to the original comments. We thus will incorporate it in the manuscript.

Furthermore, the book "The adaptive geometry of trees" by Henry Horn, which Referee #1 recommended, arrived from interlibrary loan. It was very interesting to read, and we will include Horn's considerations of his theory in the context of neighborhood and diversity in our discussion.

336, 2019.

---

## Author Response (AR1)

**Detailed response of the authors to the Referee comments**

hess-2019-336: "Neighbourhood and stand structure affect stemflow generation in a heterogeneous deciduous temperate forest" by J. C. Metzger et al.

Dear Natalie Orlowski,

Thank you for handling our manuscript.

The following is a complete list of the responses to all the comments by the reviewers. The current version of the manuscript with highlighted changes was uploaded separately. There are some very few additional changes in the current manuscript with corrected typos. We believe the review has further improved the manuscript and we hope you find it ready for publication in the current state.

Best regards,

Johanna Metzger and co-authors

**Referee #1**

Referee #1:
General Comments
The article by Metzger et al. was a pleasure to read. I found the paper to be novel, well written, well referenced, methodologically (field and statistical analyses) sound, and of wide appeal to a global audience. The tables and figures were also clear and readily interpretable. Therefore, I recommend publication following minor revision. My comments mostly regard readability and some minor technical comments that the authors should address.
Specific Comments
1) The statistical linear mixed effects models was nicely done. However, to make the analysis readily accessible to the reader, I recommend a flow chart walking one through the analysis. Such a figure in the Methods section would really clarify the sequence of the analysis and serve the reader.
*Response: We included a workflow of the model selection in the manuscript (new Fig. 2). It was already presented to and approved by Referee #1 in the discussion process.*
2) The last sentence of the abstract is a bit awkward to read and should be rephrased.
*Response: Rephrased (p. 1, ll. 29-31 of the manuscript with highlighted changes).*
3) Section 4.1 might benefit from the incorporation of Levia et al 2010 (https://doi.org/10.1016/j.jhydrol.2009.10.028) as this paper specifically discusses tree size in relation to the development of stemflow at the intra-storm scale, even showing the delay in stemflow production by larger trees of the same species due to differences in bark water storage capacity.
*Response: Yes. Incorporated (p.10, l. 28 and p. 11, ll.1-2 and p. 17, ll. 20-21 of the manuscript with highlighted changes).*
4) Page 10, line 30: I believe that the authors are referring to the following reference: https://doi.org/10.1016/j.atmosenv.2011.03.022 and not the Levia et al (2011) reference currently cited. Please check.
*Response: Yes, the Referee is right: the two Levia et al. (2011) got mixed up here, we corrected that (p. 17, ll. 22-25 of the manuscript with highlighted changes). We apologize and thank for the attentive reading!*

5) Page 11, line 27: please delete "repeatedly"

*Response: Yes, that didn't make sense. Deleted (p. 12, l. 5 of the manuscript with highlighted changes).*

6) Section 4.2.2: I really found this section interesting and I liked how some of the forest ecology aspects were brought in. In this light, I think that the authors would find the 1971 classic book "The Adaptive Geometry of Trees" by Henry Horn (Princeton University Press) of great interest. It is their choice as to whether they wish to add this text to their paper but I believe that they would find it both informative and useful.

*Response: Indeed, reading this small book was a very interesting addition to our forest ecological knowledge. We added a paragraph to the discussion (p. 11, ll. 22-28; p. 13 l. 17 and p. 17, l. 6 of the manuscript with highlighted changes).*

**Referee #2**

Referee #2:

General Comments

My comments and observations are inline with the comments posted by reviewer #1. The article by Metzger et al. was very well done. The methodology was sound and the paper sheds new and interesting light on factors influencing stemflow production that will appeal to a global audience. I also recommend publication following minor revision.

Specific Comments

My specific comments are as follows:

1) Page 2, Line 4: Perhaps of starting the sentence with "They" the authors could say "These pathways. . ."

*Response: Changed (p. 2, l. 4 of the manuscript with highlighted changes).*

2) Page 2, Lines 9 – 11. Perhaps the reference of Carlyle-Moses et al. 2018 (Advances in Water Resources) could be added here.

*Response: Done (p. 2, l. 11 of the manuscript with highlighted changes).*

3) Page 2 and 3, Lines 32 – 3: Crown projection area has also been found to be very, very important and probably should be included here.

*Response: It is absolutely true that crown projection area has been found to be very important for stemflow production, and that it therefore should be mentioned here more explicitly. The text passage Referee #2 is referring to discusses tree traits not related to tree size as impacting factors of stemflow variation. The authors think that crown projection area would rather count among the metrics for tree size which have been found to be important for stemflow. Therefore, we would like to add it a bit earlier (p.2, ll. 30-31 of the manuscript with highlighted changes, including a fitting reference). We hope that this is in consent with Referee #2.*

4) Page 7, line 21: Perhaps state: ". . .the largest 30 % of events"

*Response: Done (p. 7, l. 21 of the manuscript with highlighted changes).*

5) Page 14, Line 17: Perhaps "individual tree stemflow" instead of "tree individual stemflow"

*Response: Changed that (p. 14, l. 29 of the manuscript with highlighted changes). Changed the same expression also on p. 5, l. 24; p. 8. l. 6; p. 32 (caption of Fig. 6) and p. 36 (caption of appendix Fig. A2) of the manuscript with highlighted changes.*

6) Table 3: should the sign ">" be "<" 0.01 in the Psf column. I believe the authors mean less than and not more than here.

*Response: Yes, of course! We are sorry for the mistake (p. 22 of the manuscript with highlighted changes).*

[revised manuscript text omitted]
. Additionally, Levia et al. (2010) observed higher delays in stemflow channelling at rainfall variation for larger trees of the same species. In either of these cases, stemflow generation depends on critical event size thresholds. This view is supported by our findings: At small events, factors shaping spatial stemflow patterns are mostly random and of low temporal stability, indicating that flow paths are not yet well established. Medium events are characterized by increased temporal

5    stability of spatial ranks, however low explained variance in the fixed effects, indicating that flow paths are only partly developed. For large events, tree traits related to water collection or channelling capability are the most important factors explaining individual tree stemflow, which indicates that flow paths are fully established. Together, these results suggest that increasingly established flow paths with increasing event size invoke spatially stable patterns of stemflow that are more related to tree attributes and less to event properties.

10   **4.2 Neighbourhood and stand properties affect stemflow**

**4.2.1 Stand structure effects largely explain subplot stemflow**

For large events on the subplot scale, all proposed stand structural parameters are significant. Subplot ID has no random effect, thus, selected stand characteristics in the fixed effects capture the stemflow generation processes on the subplot scale well, including also those unexplained morphological factors which are hidden in the tree ID on the individual tree scale. Also, the

15   subplot scale model explains more variance compared to the tree individual model.

For large events on the tree individual scale, in neighbourhood effects only appeared only as trends, which may have been related to different neighbourhood variables, such as number of trees vs. basal area, working in different directions. However, the subplot models reveal that those neighbourhood effects identified at the individual tree level act in the same way at the subplot level: The number of trees still increases the stemflow on the subplot level, while basal area reduces it. This shows

20   that a tree's neighbours systematically affect its stemflow and that those patterns are not cancelling each other out when considering community stemflow at the subplot scale. Moreover, this suggests that the tree morphologic properties hidden in tree ID on the tree individual scale are actually shaped associated by with stand and neighbourhood dynamics.

It is not surprising that stand structure in a recruiting forest is organised in a patchy fashion. Because of the enormous competition for light, a climax forest cannot regenerate but in spatial and temporal niches, e.g due to the invasion of clearings

25   due to death of mature trees or other environmental heterogeneities (Horn, 1971). In consequence, regeneration patterns in an undisturbed forest, like the one observed here, organize into a juxtaposition of patches with different stand age, species composition and structure. This structural mosaic is also obvious from the variation in our subplot scale stand metrics and our data suggest that it propagates to ecohydrological functioning.

[revised manuscript text omitted]
Levia, D. F., Carlyle-Moses, D., and Tanaka, T.: Forest hydrology and biogeochemistry: synthesis of past research and future directions, Springer Science & Business Media, 2011. Levia, D. F., Van Stan, J. T., Siegert, C. M., Inamdar, S. P., Mitchell, M. J., Mage, S. M., and McHale, P. J.: Atmospheric deposition and corresponding variability of stemflow chemistry across temporal scales in a mid-Atlantic broadleaved deciduous forest, Atmos. Environ., 45, 3046-3054, 2011.

[revised manuscript text omitted]

Figure 87: (Left) Predicted stemflow per subplot using the subplot linear mixed effects model in relation to observed values for the large event class, (right) stemflow sums per subplot predicted by the individual tree linear mixed effects model in relation to observed values. The right panel shows additionally the predicted values when excluding the tree ID random effect from the individual tree model. Dashed lines give the 1-to-1-line, continuous lines show the linear regressions, equations are given in the graph.

[Figure]

**Figure A1: Histograms of stand properties on the whole 1-ha-plot (left, n = 581) and the eleven 100-m²-subplots on which stemflow was measured (right, n = 65).**

[Figure]

**Figure A2: Distributions and correlations of variables included as fixed effects in the linear mixed effect models of  individual tree stemflow. Abbreviations: t_dbh: Tree DBH, t_h: Tree height, n_not: Number of trees in the neighborhood, n_ba: Neighborhood basal area, n_rh: Neighborhood relative height. DBH: Diameter at breast height.**

[Figure]

**Figure A3: Distributions and correlations of variables included as fixed effects in the linear mixed effect models of subplot stemflow. Abbreviations: sp_not: Number of trees in the subplot, sp_nospec: Number of species in the subplot, sp_simp: Simpon's diversity index of the subplot, sp_ba: Basal area of the subplot, sp_dbhmax: DBH of the biggest tree on the subplot, sp_shi: Size heterogeneity index of the subplot, sp_lai: Subplot LAI. DBH: Diameter at breast height, LAI: leaf area index.**